# Hemispheric asymmetry in recent stratospheric age of air changes

Kimberlee Dubé[1], Susann Tegtmeier[1], Felix Ploeger[2], and Kaley A. Walker[3]

[1]Institute of Space and Atmospheric Studies, University of Saskatchewan, Saskatoon, SK, Canada
[2]Forschungszentrum Jülich, Jülich, Germany
[3]Department of Physics, University of Toronto, Toronto, ON, Canada

**Correspondence:** Kimberlee Dubé (kimberlee.dube@usask.ca)

**Abstract.** Many stratospheric trace gases, including $O_3$, $HCl$, and $NO_y$, have opposing trends in the Southern Hemisphere (SH) compared to the Northern Hemisphere (NH) during the last two decades. Some of this difference is due to hemispherically asymmetric changes in the rate of transport by the Brewer-Dobson Circulation (BDC), and some is due to ozone depletion and recovery. The mean Age of Air (AoA) is a common proxy for the transport rate by the BDC in models, however it cannot be directly measured. We use observations from the Atmospheric Chemistry Experiment Fourier Transform Spectrometer (ACE-FTS) along with results from the Chemical Lagrangian Model of the Stratosphere (CLaMS) to derive AoA anomalies and AoA trends. The AoA is derived using observations of $N_2O$, $CH_4$, and $CFC-12$, all long-lived trace gases with tropospheric sources. We also consider CLaMS simulations driven with four different reanalyses (ERA5, ERA-Interim, JRA-55, MERRA-2). We find that, irrespective of which trace gas or reanalysis is used, air in the NH aged by up to 0.3 years/decade relative to the SH over 2004–2017. The maximum hemispheric difference in aging occurs in the middle stratosphere, near 30 hPa ($\sim$24 km). We also show that the aging rate in the NH becomes smaller when the analysis is extended to 2021. The observed aging in the NH middle stratosphere contradicts model predictions of a decrease in stratospheric AoA in response to rising atmospheric greenhouse gas levels. However, the smaller aging rate during 2004–2021 compared to 2004–2017 provides some evidence that the NH aging is impacted by decadal variability and the limited length of the observation period.

## 1 Introduction

Anthropogenic greenhouse gas emissions are altering the temperature, dynamics, and composition of the atmosphere at unprecedented rates. Observations from the past several decades show a warming troposphere and a cooling stratosphere (e.g., Steiner et al., 2020), and the consequences of these changes: tropical expansion (e.g., Seidel et al., 2008), an increased tropopause height (e.g., Vallis et al., 2015), and an altered Brewer-Dobson Circulation (BDC, e.g., Stiller et al., 2017; Bönisch et al., 2011; Fu et al., 2019), amongst others. At the same time, Antarctic ozone recovery in the 21st century following a successful implementation of the Montreal Protocol is warming the Southern hemisphere (SH), and further changing the BDC in a way that is not symmetric between the hemispheres (e.g., Abalos et al., 2019). An understanding of these structural changes to the stratosphere is needed in order to make sense of recent stratospheric trace gas trends. HCl (Mahieu et al., 2014; Strahan et al., 2020), $NO_x$ (Galytska et al., 2019; Dubé et al., 2020), $F_y$ (Prignon et al., 2021), and $O_3$ (Ball et al., 2018; Bognar et al., 2022) observations all have opposing trends in the SH relative to the Northern hemisphere (NH) in the years since $\sim$2000.

These trends cannot be explained solely by changes in source gas emissions and there is evidence that they are due to changes in transport by the BDC (e.g., Mahieu et al., 2014; Dubé et al., 2023; Ploeger and Garny, 2022; Prignon et al., 2021; Han et al., 2019).

The BDC is the mechanism for long-range mass transport within the stratosphere. It consists of upwelling in the tropics, poleward transport, and downwelling at mid and high latitudes, as well as quasi-horizontal two-way mixing (Butchart, 2014). Transport within the BDC is commonly quantified by the age of air, which defines the length of time that an air parcel requires to travel from either the surface or the tropopause to some location in the stratosphere (Hall and Plumb, 1994; Waugh and Hall, 2002). The age of air is a distribution (or spectrum) of all possible transit times to a given location. The first moment of the age of air spectrum, the mean stratospheric age of air (AoA), is a metric used to represent the mean transit time of an air parcel to a given point in the stratosphere.

Climate models predict that rising atmospheric greenhouse gas concentrations will lead to an accelerated BDC, and therefore younger air throughout the stratosphere (Li et al., 2018; Abalos et al., 2021). Before $\sim$2000 this effect was compounded by ozone depletion, which lead to an even greater decrease in AoA in the SH relative to the NH below 10 hPa (Abalos et al., 2019; Polvani et al., 2018). The recovery of the ozone layer in more recent years is expected to produce an opposite effect, resulting in air that is older in the SH relative to NH, which will counteract some of the BDC acceleration from greenhouse gases (Polvani et al., 2018).

Reanalyses are datasets with high vertical and horizontal resolution that are generated by assimilation systems which combine global weather forecast models and input observations from many sources to provide a physically consistent estimate of the past atmospheric state (Fujiwara et al., 2017). A Chemical Transport Model (CTM) can be used to calculate AoA using reanalysis fields as input (e.g., Monge-Sanz et al., 2007). Monge-Sanz et al. (2012) were the first to find an increase in AoA in the NH relative to the SH, using the TOMCAT/SLIMCAT CTM (Chipperfield, 2006) driven with ERA-Interim (Dee et al., 2011). Several studies have since shown that AoA trends derived from reanalyses are very different depending upon which reanalysis is considered, regardless of the CTM that is used (Chabrillat et al., 2018; Ploeger and Garny, 2022; Monge-Sanz et al., 2022). However, Ploeger and Garny (2022) found that the hemispheric difference in the AoA trend (the trend in the NH-SH difference) is much more similar amongst the reanalyses than the individual trends in each hemisphere: All four reanalyses that were considered show an aging of the NH relative to the SH over 2005–2017, which is not consistent with the expected response to ozone recovery.

It is not possible to directly measure the AoA, however it can be approximated from observations of a long-lived trace gas with a linearly varying tropospheric source. The most common choices are sulphur hexafluoride ($SF_6$) and carbon dioxide ($CO_2$), which are inert in much of the stratosphere, and have well defined tropospheric trends (Waugh and Hall, 2002; Stiller et al., 2008). AoA derived from Michelson Interferometer for Passive Atmospheric Sounding (MIPAS, Fischer et al., 2008) $SF_6$ observations has a linear change over 2002–2012 that is negative (air getting younger) in the SH and positive (air getting older) in the NH, between 20 and 30 km (Haenel et al., 2015). For the longer time period of 1975–2016, Engel et al. (2017) did not observe a significant AoA trend in the NH based on balloon measurements of $CO_2$ and $SF_6$ between 30 hPa and 5 hPa. These results from Engel et al. (2017) are largely inconsistent with the modelled AoA trends presented in Abalos et al. (2021).

Garcia et al. (2011) showed that sparse spatial and temporal sampling of observations, as is the case in the dataset used by Engel et al. (2017), results in high uncertainties in observational AoA trends, and contributes to differences between observed and modelled AoA trends.

Attempts to derive AoA from observations of other long-lived trace gases have also been made. Linz et al. (2017) derived AoA from Global OZone Chemistry And Related trace gas Data records for the Stratosphere (GOZCARDS) $N_2O$ observations at 20 km. Their method relied on the relationship between AoA and $N_2O$, where AoA is derived from aircraft and balloon observations of $CO_2$. The AoA from $N_2O$ was found to be younger than the AoA from MIPAS $SF_6$. Linz et al. (2017) did not compare trends in the two AoA versions. Strahan et al. (2020) considered other trace gases, using HCl and $HNO_3$ observations from the Network for the Detection of Atmospheric Composition Change (NDACC) to derive the AoA trend. They found that air in the SH became 1 month/decade younger relative to air in the NH over 1994–2018. The longer time period prevents direct comparison, but the pattern of an aging NH relative to the SH is consistent with MIPAS and with results from CTMs driven with various reanalyses (Monge-Sanz et al., 2012; Diallo et al., 2012; Haenel et al., 2015; Ploeger and Garny, 2022).

The limited time period of the MIPAS observations, the limited altitude range of the Linz et al. (2017) $N_2O$-based AoA, and the sparseness of the balloon measurements used by Engel et al. (2017) motivate the creation of a new observational based AoA dataset that can be used to validate model results. Strahan et al. (2020) provided some new information by deriving AoA trends from HCl and $HNO_3$, however they used column measurements and so could not compute the AoA as a function of altitude. Their analysis was also limited to the latitudinal resolution of the NDACC stations, and therefore focused on the interhemispheric difference. Our goal in this study is to build on the results of Strahan et al. (2020) by applying their method for calculating the AoA trend to observations from the Atmospheric Chemistry Experiment - Fourier Transform Spectrometer (ACE-FTS, Bernath et al., 2005). By using ACE-FTS observations it is possible to calculate the AoA anomaly and the AoA trend as a function of altitude and latitude. The downside compared to using NDACC observations is that we cannot consider AoA before ACE-FTS began operations in 2004.

The AoA is calculated using profiles of $N_2O$, $CH_4$, and $CFC-12$ retrieved from ACE-FTS observations, along with the relationship between these trace gases and AoA in the Chemical Lagrangian Model of the Stratosphere (CLaMS, McKenna et al., 2002). These gases were chosen as they are long-lived tracers that provide a good representation of dynamic variability in the lower and middle stratosphere. We consider four different sets of CLaMS results, based on inputs from four different reanalyses. The ACE-FTS observations and CLaMS are described in Section 2. Section 3 contains the method for deriving AoA and its trend, along with the corresponding results. A conclusion is given in Section 4.

## 2 Observations and Model

ACE-FTS is an infrared Fourier transform spectrometer that measures from 750–4400 $cm^{-1}$ (Bernath et al., 2005; Boone et al., 2005). It has been observing the atmosphere from a high inclination orbit on the SCISAT satellite since February 2004. ACE-FTS uses a solar occultation viewing geometry to make approximately 30 atmospheric transmission profile measurements each day, ~15 at sunrise and ~15 at sunset. Vertical volume mixing ratio (VMR) profiles of over 40 trace gas species are retrieved

from ACE-FTS measurements. We use $N_2O$, $CH_4$, and $CFC-12$ VMR profiles from version 4.2 of the retrieval, described in
Boone et al. (2020). The VMR profiles are filtered according to the data quality flags developed by Sheese et al. (2015) before doing any analysis.

Simulations of $N_2O$, $CH_4$, $CFC-12$, and AoA profiles from CLaMS, a Lagrangian chemistry transport model, are also considered. The CLaMS transport scheme includes 3D air parcel trajectory calculations and parameterizations for small-scale atmospheric mixing. AoA is modelled in CLaMS by tracking the time it takes for an inert tracer with a linearly increasing
source that is released at the surface to reach a certain location in the stratosphere (Ploeger and Birner, 2016; Ploeger et al., 2021). CLaMS uses a simplified chemistry scheme to model long-lived trace gases (Pommrich et al., 2014).

Reanalysis horizontal winds and diabatic heating rates are provided as input to CLaMS. Four different reanalyses are considered to test the sensitivity of the modelled AoA and gas trends to the input values: The European Centre for Medium-Range Weather Forecasts (ECMWF) interim reanalysis, ERA-Interim (Dee et al., 2011), the ECMWF fifth generation reanalysis,
ERA5 (Hersbach et al., 2020), the Modern-Era Retrospective analysis for Research and Applications Version 2, MERRA-2 (Gelaro et al., 2017), and the Japanese 55-year Reanalysis, JRA-55 (Kobayashi et al., 2015).

A detailed discussion of the AoA trends for each reanalysis can be found in Ploeger and Garny (2022). The same CLaMS simulations for each reanalysis are used here. The simulations all begin in 1979 (except the MERRA2 simulation which begins in 1980 due to data availability) but the end years of the simulations based on each reanalysis are different, depending on
the availability of the reanalysis results. The same initial conditions for trace gases and age of air for all four simulations are taken from climatological data (Pommrich et al., 2014). Section 3 focuses on results for 2004–2017 as this is the time period for which both ACE-FTS observations and results from all four reanalyses were available when the model simulations were performed. Results for 2004–2021 are also discussed in the case of ERA5, for which longer simulations are available.

We calculate trends in both the AoA interhemispheric difference (SH−NH) and the latitudinally resolved AoA. As the focus
is on trends due solely to circulation changes, trends caused by changes in surface emissions of the trace gases need to be accounted for. For the interhemispheric difference we assume that the effect of surface emissions on the trend is the same in both hemispheres and so cancels out, following Strahan et al. (2020). The latitudinally resolved results are based solely on $N_2O$, and not also $CH_4$ and $CFC-12$, as $N_2O$ has a well defined tropospheric source trend that has been constant for several decades (Laube et al., 2022; Canadell et al., 2021). This allows us to remove the effect of $N_2O$ emissions on the stratospheric $N_2O$ to get
the trend that is only due to changes in stratospheric transport. To do so, a time series of $N_2O$ surface concentrations is required: we use global monthly-mean surface $N_2O$ measurements from the National Oceanic and Atmospheric Administration Global Monitoring Laboratory (NOAA/GML) Halocarbons and other Atmospheric Trace Species (HATS) flask sampling programme (NOAA/GML, 2022). This dataset combines $N_2O$ observations from 13 stations across the globe.

## 3 Method and Results

We start by comparing the ACE-FTS VMRs and the CLaMS results. This requires interpolating the ACE-FTS VMR profiles from altitude to pressure levels using the pressure profile that is retrieved from each ACE-FTS occultation, and interpolating the

CLaMS output to the locations and times representative of the whole ACE-FTS profiles, using their locations and times at 30 km. The individual ACE-FTS and CLaMS profiles are then used to determine the deseasonalized monthly zonal mean (MZM) anomalies at each pressure level. This is calculated by first taking the mean of all data points for a month within a 10 degree latitude band, and then subtracting the multi-annual mean for a given month of the year from all values for that month (i.e. the mean January value is subtracted from all January values). We focus on mid-latitudes to avoid the effect of polar chemistry, so in all subsequent discussion the SH is defined as 50°S–20°S and the NH is defined as 20°N–50°N. Figure 1 shows the 26.1 hPa MZM anomaly time series of $N_2O$ and AoA for ACE-FTS observations ($N_2O$ only) and for the four CLaMS simulations. A similar plot showing $CH_4$ and $CFC-12$ is included in the Appendix, Figure A1. The gaps in the time series are due to the ACE-FTS sampling pattern.

Both Figure 1 and appendix Figure A1 show that the MZM ACE-FTS VMR anomalies and the CLaMS VMR anomalies from all simulations have very similar variability on a monthly time scale. The correlation of the NH ACE-FTS $N_2O$ and the NH CLaMS $N_2O$ is greater than 0.5 at all levels between 100 hPa and 1 hPa, no matter which CLaMS run is considered. Similar correlation levels also hold in the SH.

Despite the similar short-term variability amongst ACE-FTS and the four model runs, there is a clear difference between the datasets when considering changes over the full 17-year period. For example, in 2006 the NH $N_2O$ anomaly in CLaMS driven with ERA-Interim at 26.1 hPa is around 50 ppbv, while the anomaly in MERRA-2 for the same year and level is closer to 10 ppbv. This large difference near the beginning of the time period can result in significantly different trends for each reanalysis. However, there is much less of a bias between ACE-FTS and the different reanalyses when considering the difference between the SH anomaly and the NH anomaly (panels E and F of Figure 1). This was previously shown by Ploeger and Garny (2022), who found that even though AoA trends are very different amongst the reanalyses, the trends in the interhemispheric difference all have a similar structure, with greater aging in the NH relative to the SH during 2005–2017. This suggests that even though the reanalyses do not have the same absolute AoA changes, they are all capturing the AoA changes in a way that is consistent between the hemispheres.

## 3.1 Trend in AoA Interhemispheric Difference

We first derive the trend in the AoA interhemispheric (SH−NH) difference as a function of pressure level, as an update to the results of Strahan et al. (2020), who only considered trends at 52 hPa. This requires assuming that the trace gas VMRs being used represent dynamical variability in the atmosphere. The level of correlation between AoA and a trace gas shows the similarity of the month to month variability. The correlation between AoA and $N_2O$, $CH_4$, and $CFC-12$ from CLaMS forced with each reanalysis is shown in the top row of Figure 2, while the bottom row of Figure 2 shows the same correlations but using ACE-FTS VMR anomaly profiles. Note that the ACE-FTS $CFC-12$ observations are only available up to ∼8 hPa.

All three trace gas VMR anomalies are strongly anti-correlated with the AoA anomalies below ∼3 hPa. At these levels the gases have long chemical lifetimes so their distributions in the stratosphere are controlled by transport processes, for which AoA is a metric. The anti-correlation occurs because all three trace gases have tropospheric sources and stratospheric sinks so older air implies the gas has had more time to be removed by chemical processes. Subsequent analysis is limited to levels

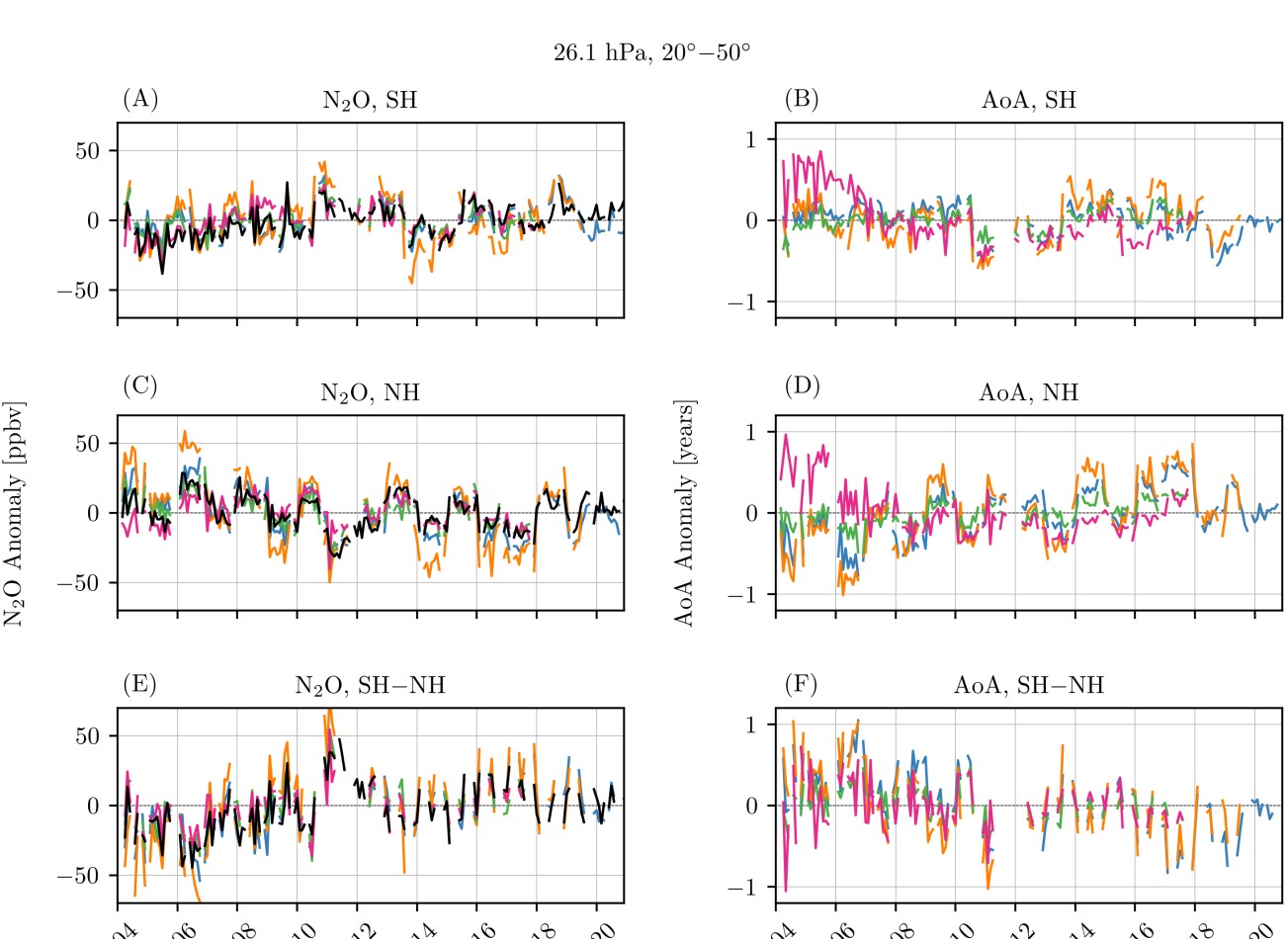

**Figure 1.** Left: Deseasonalized monthly zonal mean anomaly time series at 26.1 hPa for ACE-FTS $N_2O$ and $N_2O$ from CLaMS driven with four different reanalyses. Right: Deseasonalized monthly zonal mean anomaly time series for AoA from CLaMS driven with four different reanalyses. Panels are divided into SH (top row), NH (centre row), and SH−NH difference (bottom row).

where all anti-correlations are less than -0.5, implying that there is a strong relationship between the AoA anomaly and the trace gas anomaly. This corresponds to levels below 3 hPa for $N_2O$ and $CH_4$, and below 10 hPa for CFC−12.

Although the AoA and the trace gas VMR profiles simulated by CLaMS are quite different depending on the input reanalysis winds and heating rates, the relationship between AoA and the gas VMRs should always be consistent as it largely depends

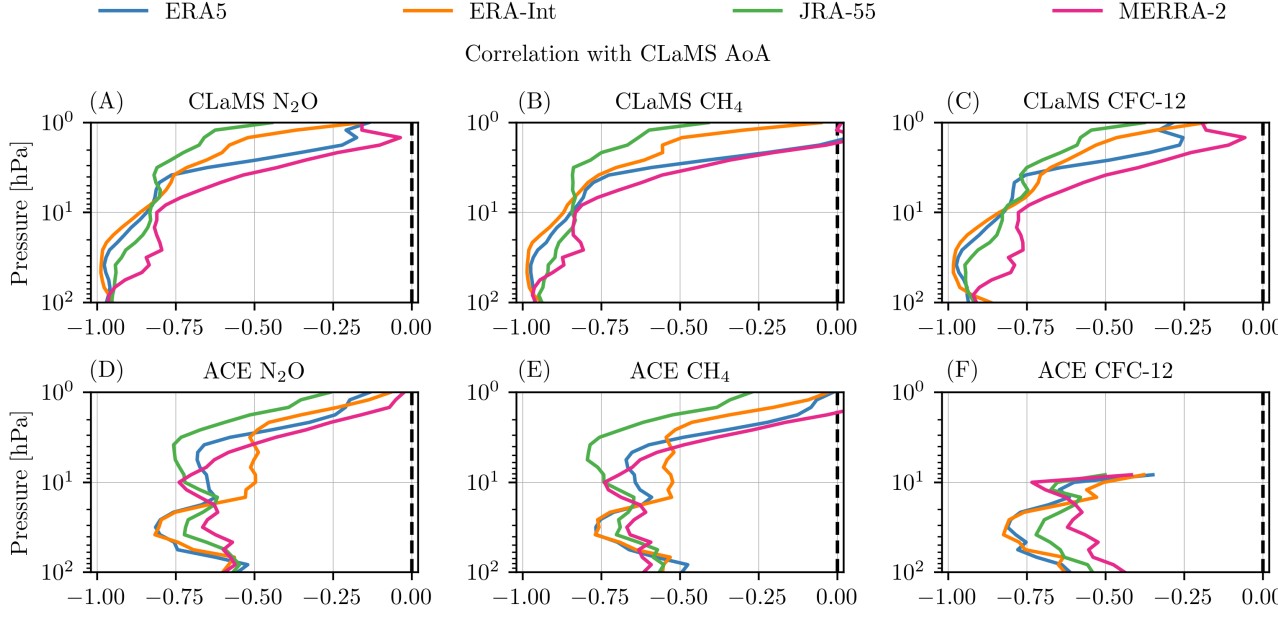

**Figure 2.** Correlation of $N_2O$, $CH_4$, and $CFC-12$ with AoA from CLaMS driven with four different reanalyses as a function of pressure level. Top row: VMR anomaly profiles from CLaMS. Bottom row: VMR anomaly profiles from ACE-FTS. All results for SH−NH difference over 2004–2017. CFC−12 observations from ACE-FTS are only available up to 8 hPa.

on the chemistry in CLaMS. This relationship, in the SH−NH difference, is shown for each trace gas and reanalysis, and for two pressure levels, in Figure 3. The three trace gases are plotted as the relative anomaly, which is the anomaly divided by the overall mean and multiplied by 100, so that it is in units of percent. The overall mean profiles for each trace gas are provided in the Appendix, Figure A2. The relative anomaly is calculated in each hemisphere separately, before taking the SH−NH difference.

As expected from the high correlations in Figure 2, each panel of Figure 3 shows a tight relationship between the AoA and the VMR relative anomalies. The trend lines in Figure 3 are computed by calculating the least-squares fit between the AoA anomaly and the trace gas relative anomaly. We call this the slope, and compute it for each pressure level, trace gas, and reanalysis. The error in the slope is the $2\sigma$ error of the least squares fit. The top row of Figure 4 shows the resulting slopes in the SH−NH difference for AoA with each of $N_2O$ (panel A), $CH_4$ (panel B), and $CFC−12$ (panel C). Although the slope for each reanalysis is not within error of the least squares fit for every other reanalysis at every altitude, the slopes have a very similar vertical structure and similar values, no matter which reanalysis is used. The slopes are not the same for each trace gas because the gases are affected by different chemical mechanisms.

To derive the ACE-FTS AoA trend the slope is multiplied by the ACE-FTS trace gas trend,

$$\text{ACE AoA Trend [years/decade]} = \frac{\text{CLaMS AoA [years]}}{\text{CLaMS Gas [\%]}} \times \text{ACE Gas Trend[\%/decade]}. \tag{1}$$

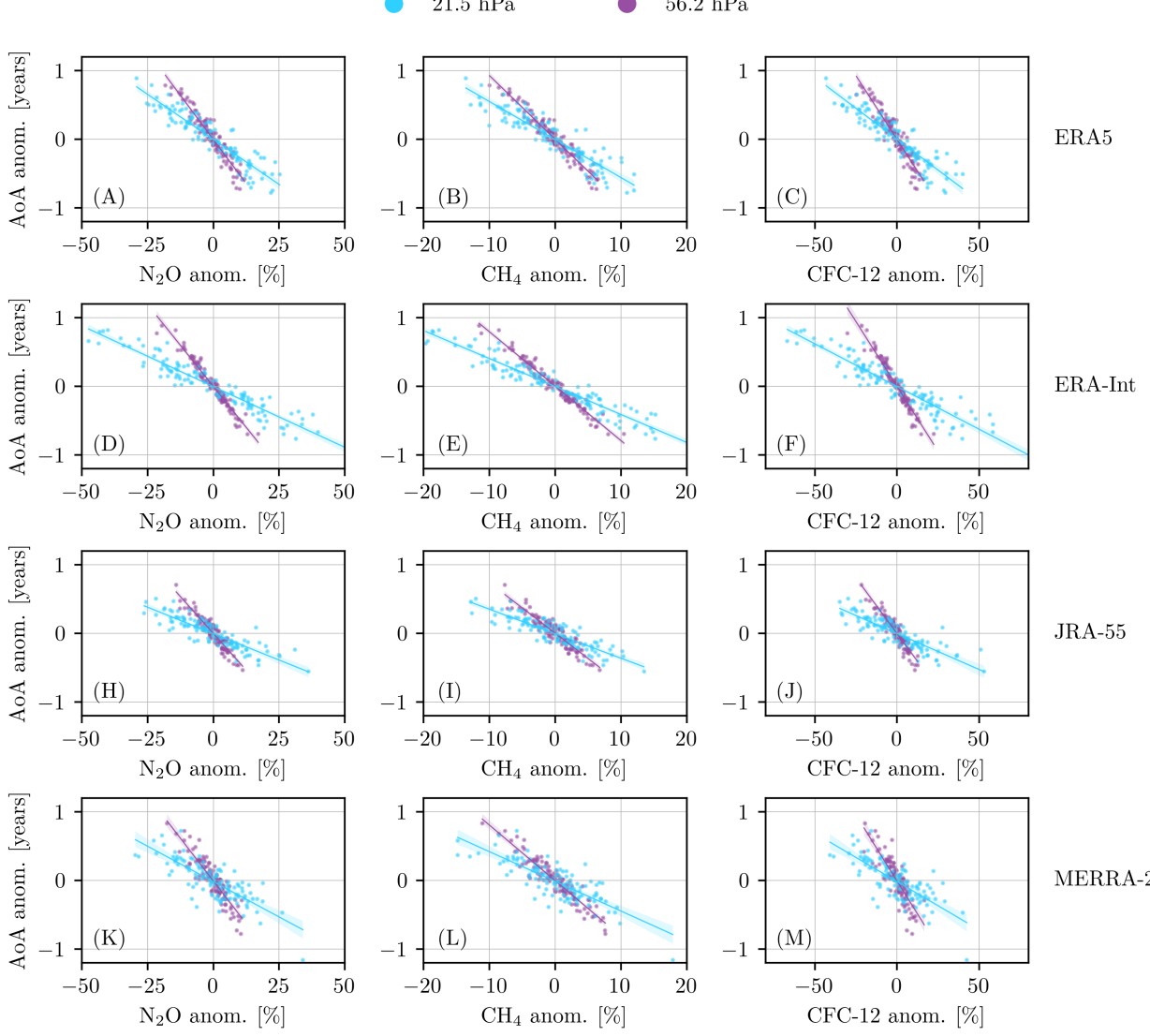

**Figure 3.** Scatter plots showing the relationship between the SH−NH difference for the AoA anomaly and relative anomalies for each of $N_2O$ (first column), $CH_4$ (second column), and CFC−12 (third column). Results are shown for two different pressure levels, and for CLaMS driven with four different reanalyses. The solid lines show the least squares fit to the data points, and the shaded regions are the $2\sigma$ error in the linear fits.

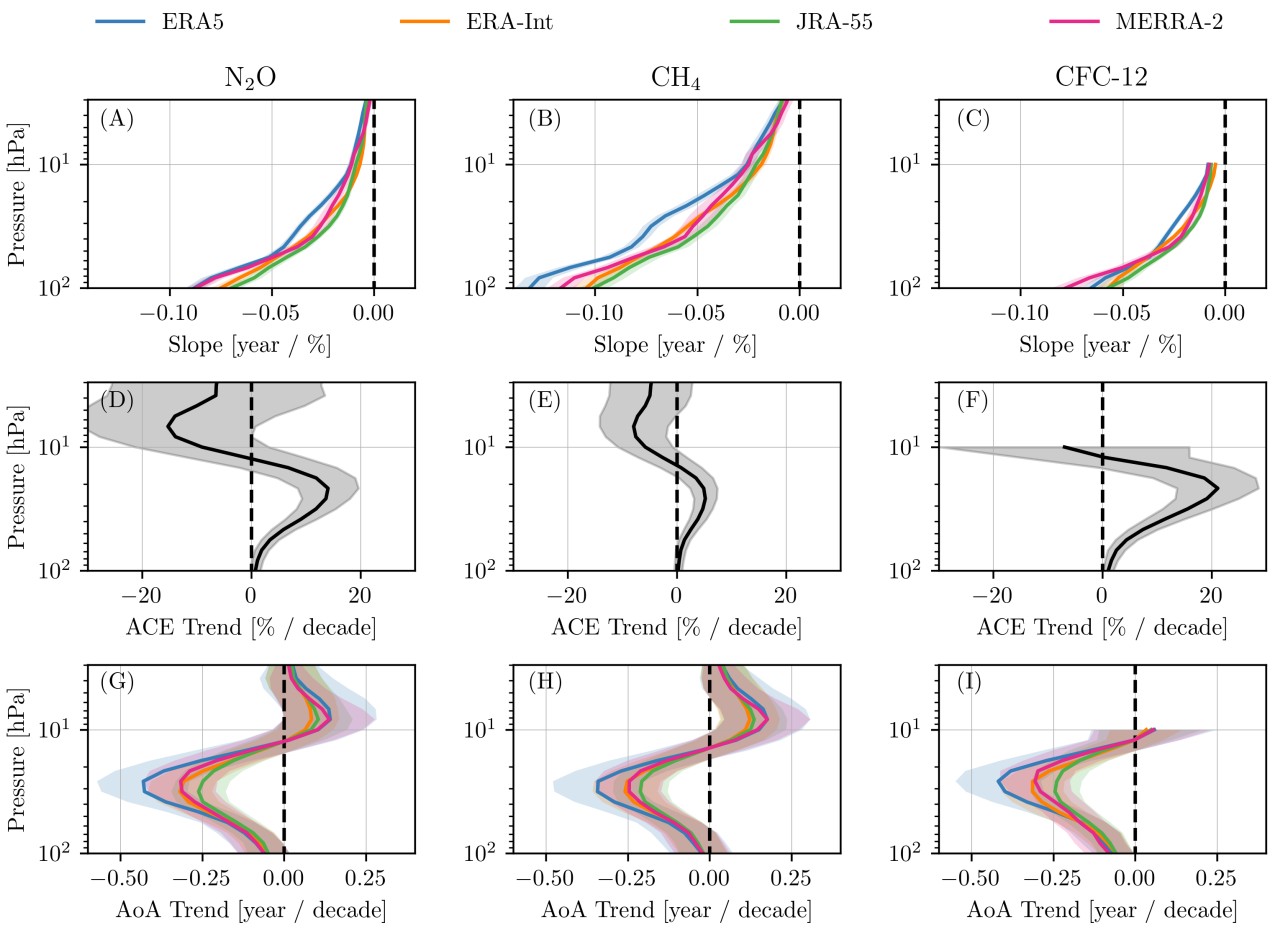

**Figure 4.** Top row: Slope from least-squares fit to CLaMS AoA anomalies and corresponding trace gas VMR relative anomalies. Centre row: Trend in ACE-FTS VMR relative anomalies. Bottom row: Derived AoA trend. All results for SH−NH difference over 2004–2017 and as a function of pressure level. Results are shown for CLaMS driven with four different reanalyses. Shaded regions denote the $2\sigma$ error.

In this case the ACE-FTS VMR anomaly trend in the interhemispheric difference is calculated with a simple linear fit to the anomalies. Panels D, E, and F of Figure 4 show these trends for each of $N_2O$, $CH_4$, and $CFC-12$, respectively. The trends are significant where the error bar does not cross the zero line, so between approximately 20 hPa and 80 hPa. At these levels there is a positive trend, so the trace gas VMR increased more in the SH relative to the NH over 2004–2017.

The resulting trends in the AoA interhemispheric difference, calculated with Equation 1, are shown in the bottom row of Figure 4. The AoA trends have an opposite structure to the trace gas trends, and show that below 10 hPa the SH became significantly younger than the NH during 2004–2017. Table 1 summarizes the AoA trends from each trace gas at several levels in the middle stratosphere, with results from all four reanalyses averaged together. A key result is that the trends in the AoA

interhemispheric differences agree within the regression error regardless of which trace gas is used, and regardless of which reanalysis is used as CLaMS input. Therefore the choice of reanalysis is not important for determining SH−NH AoA trends with this method, and neither are any specifics of the ACE-FTS observations/retrievals that may have different impacts on the three long-lived trace gases that are considered.

**Table 1.** Trends in AoA interhemispheric difference (SH−NH) for 2004–2017. Values are the mean AoA trend derived from ACE-FTS and four CLaMS runs, each using input from a different reanalysis. Trend units are year/decade. Errors are the $2\sigma$ uncertainty.

| Level [hPa] | AoA from $N_2O$ [yr./dec.] | AoA from $CH_4$ [yr./dec.] | AoA from CFC−12 [yr./dec.] |
|:-----------:|:--------------------------:|:---------------------------:|:---------------------------:|
| 21.5        | -0.28±0.12                 | -0.21±0.12                  | -0.29±0.11                  |
| 31.6        | -0.33±0.11                 | -0.27±0.10                  | -0.31±0.10                  |
| 46.4        | -0.23±0.09                 | -0.17±0.09                  | -0.22±0.08                  |
| 56.2        | -0.16±0.08                 | -0.11±0.09                  | -0.16±0.07                  |

### 3.2 Global AoA Time Series and Trend Analysis

We now extend the analysis by creating a latitudinally and vertically resolved AoA anomaly time series and using this to better understand the variability in AoA in each hemisphere individually over the past two decades. One complication when deriving AoA as a function of latitude from trace gas observations is that the effect of surface emissions on the gas trend is not automatically removed (see Section 2). To account for this it is necessary to remove the surface emission trend from the trace gas observations before calculating the AoA. This is only possible for $N_2O$, as $N_2O$ has a relatively constant emission trend since measurements began in 1977 (Canadell et al., 2021). This means that the emission driven contribution to the stratospheric $N_2O$ trends is constant throughout the stratosphere and not modified by variations in transit time. This is not true for $CH_4$ and CFC−12 as their emission rates change significantly depending on the time period considered (Laube et al., 2022; Canadell et al., 2021).

The effect of surface emissions is removed from the ACE-FTS $N_2O$ by subtracting the linear trend in the surface data from the relative anomaly of the ACE-FTS observations in each bin (the same method described in Dubé et al. (2023)). These anomalies are then multiplied by the least squares fit of CLaMS AoA to $N_2O$ (the slope) to get the ACE-FTS AoA anomaly,

$$\text{ACE AoA [years]} = \frac{\text{CLaMS AoA [years]}}{\text{CLaMS Gas [\%]}} \times \text{ACE Gas [\%]}. \tag{2}$$

The slope is calculated in the same way as in Section 3.1, but in this case the slope term is different for each pressure and latitude bin. Figure 5 shows that the slopes have the same latitudinal and vertical structure no matter which reanalysis is used in the CLaMS runs. The slope is most negative at lower levels and in the tropics.

The resulting AoA anomaly time series for four sample latitude/altitude bins are shown in Figure 6. Note that results from CLaMS run with ERA5 are available to the end of 2021. The AoA anomalies are very similar for all four reanalyses, and the

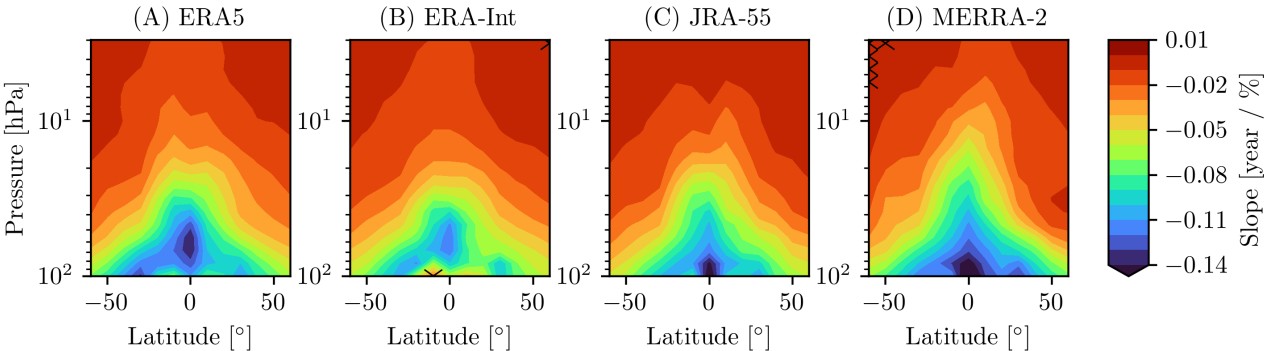

**Figure 5.** Slope from linear fit to CLaMS AoA and $N_2O$ for 2004–2017. Results are shown for CLaMS driven with four different reanalyses. Hatching denotes statistically insignificant slopes at the $2\sigma$ level (only MERRA-2 and ERA-Int have hatching, and very few bins are hatched. The hatched bins are at the highest latitudes and pressure levels).

short-term variability in the AoA from all four reanalyses is identical since all cases are a scaled version of the ACE-FTS $N_2O$ relative anomaly and the scale factors do not vary in time. One source of variability that stands out, particularly in panel A, is the quasi-biennial oscillation (QBO, Wallace et al., 1993), an oscillation in the tropical zonal winds that has an approximately 2-year period. The QBO phase has previously been shown to affect the age of air spectrum by Ploeger and Birner (2016). To

215 account for variability from the QBO, as well as possible variability from the El-Niño Southern Oscillation and the 11-year solar cycle, we calculate trends in the latitudinally resolved AoA using a multiple linear regression model (MLR).

The MLR model is defined as

$$AoA(t) = \beta + \beta_{trend} \times linear(t) + \beta^{(2)}_{qboa} \times QBO_a(t) + \beta^{(2)}_{qbob} \times QBO_b(t) +$$
$$\beta_{enso} \times ENSO(t) + \beta_{solar} \times F10.7(t) + R(t). \tag{3}$$

Each $\beta_i$ defines a regression coefficient, with the superscripts defining the highest seasonal harmonic included for that term. Harmonics are included for the QBO predictors, $QBO_a(t)$ and $QBO_b(t)$, to account for coupling between the QBO and the seasonal cycle. There is no need to include regression terms for annual oscillations as the AoA anomaly is deseasonalized. $\beta_{trend}$ is the AoA trend in units of years/decade, $F10.7(t)$ is the solar flux at 10.7 cm, $ENSO(t)$ is the El-Niño Southern Oscillation index, $QBO_a(t)$ and $QBO_b(t)$ are the first two principal components of the monthly mean zonal winds measured

in Singapore, and $R(t)$ is the residual. Further details on the regression model and the proxy data sources are provided in Damadeo et al. (2022).

The AoA trends over 2004–2017 are compared for each reanalysis in the bottom row of Figure 7. As expected, there is a significant hemispheric asymmetry in the trends. Below 10 hPa, most of this asymmetry is coming from aging of up to 0.4 years/decade in the NH. The AoA trend in the SH is largely insignificant, except between 10 hPa and 30 hPa at latitudes larger

than $40°S$, where the air is getting younger by up to 0.2 years/decade. Above 10 hPa the NH is getting younger relative to the SH. As expected, given the consistency of the AoA time series derived from each reanalysis, the AoA trends are very similar

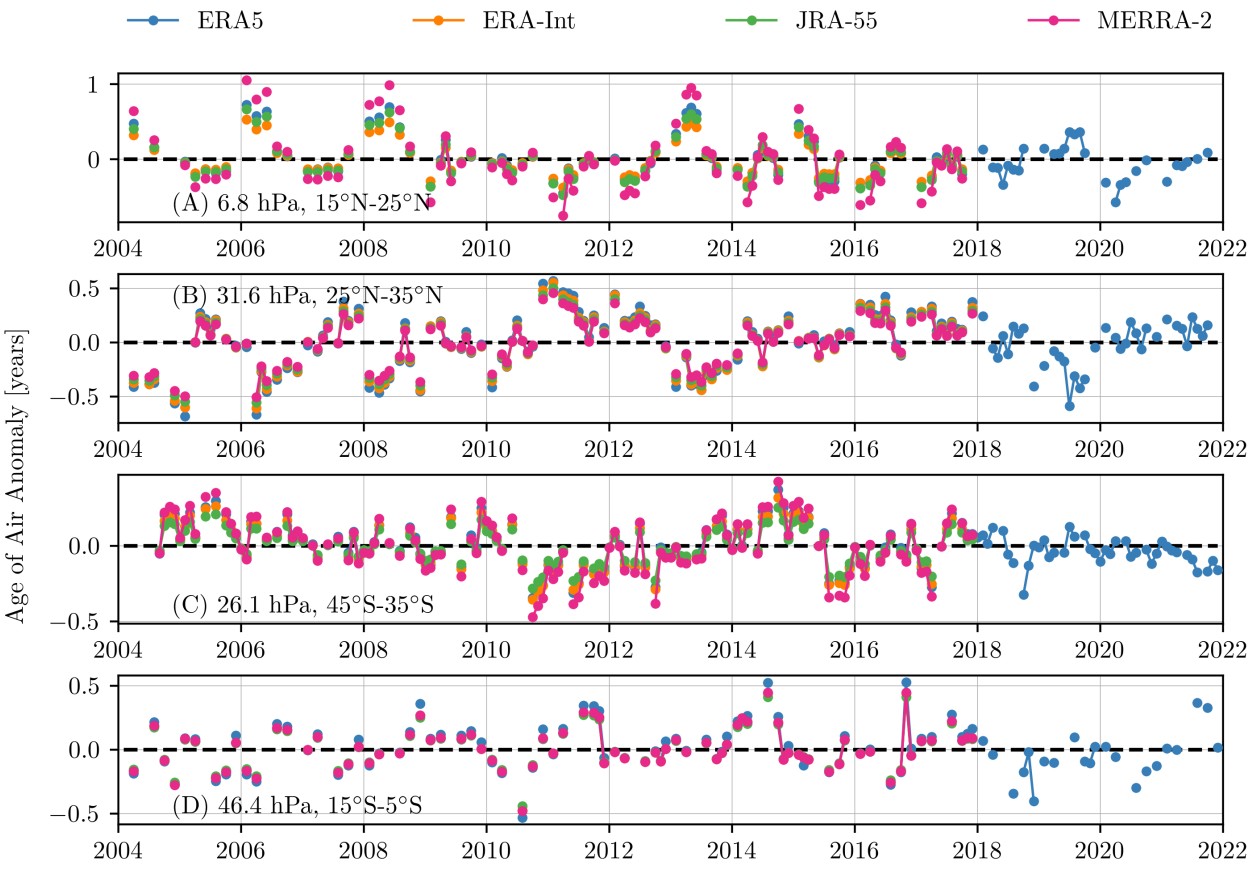

**Figure 6.** AoA anomaly for four latitude and altitude bins. The AoA anomaly is derived using ACE-FTS retrievals of $N_2O$ and the relationship between AoA and $N_2O$ in CLaMS driven with four different reanalyses

in all four cases, although ERA5 has slightly more NH aging below 10 hPa. The region of highest positive trend in the NH corresponds to panel B in Figure 6, and the region of most negative trend in the SH corresponds to panel C.

The top row of Figure 7 shows the AoA trends in the CLaMS simulations driven with four different reanalyses, when no additional information from ACE-FTS observations is incorporated. These are the same results presented in Figure 1 of Ploeger and Garny (2022). While trends based on all four reanalyses are greater in the NH than the SH, there are significant differences in the values, and even the signs, of the trends. MERRA-2 has air getting younger throughout much of the stratosphere, while JRA-55 has air getting older at most latitude and pressure levels. ERA5 and ERA-Int both have the same increasing AoA in the NH and decreasing AoA in the SH below 10 hPa that we see in the ACE-FTS observations, but NH aging is much greater in the reanalyses than in ACE-FTS. The differences between AoA trends derived from a CTM forced with different reanalyses are also apparent in other time periods and for CTMs besides CLaMS, as demonstrated by Monge-Sanz et al. (2022). The dissimilarity between the CLaMS-only AoA trends for each reanalysis, compared to the similarity of the AoA trends derived

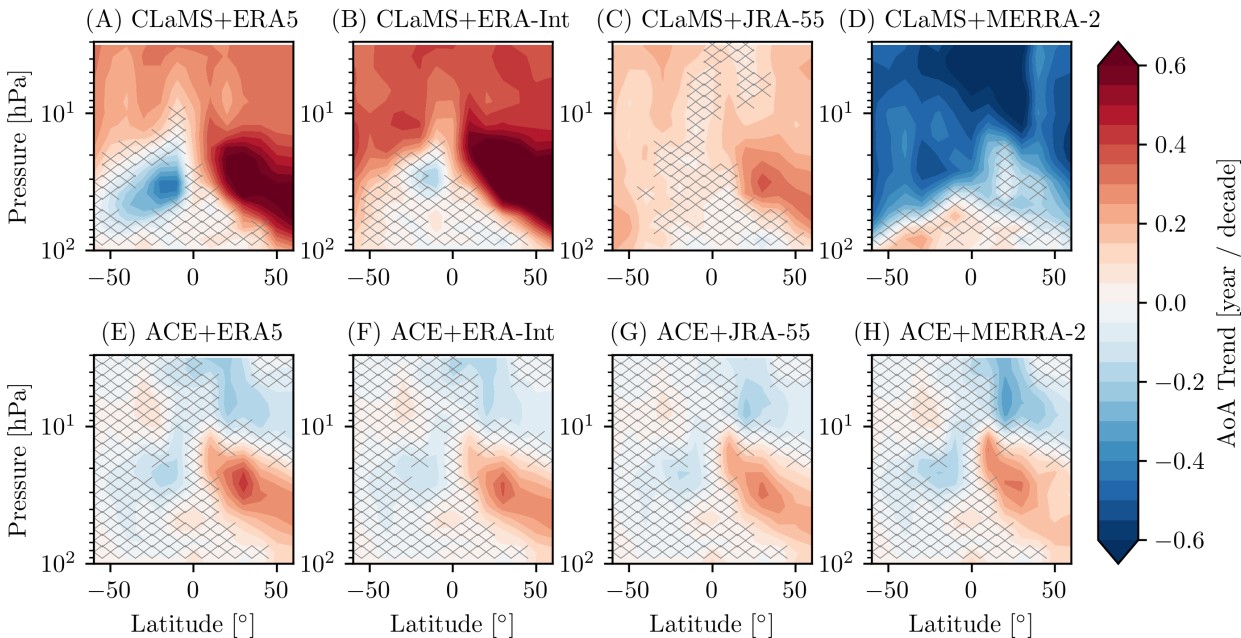

**Figure 7.** Top row: AoA trends from CLaMS driven with four different reanalyses. Bottom row: AoA trends derived using ACE-FTS retrievals of $N_2O$ and the relationship between AoA and $N_2O$ in CLaMS driven with four different reanalyses All AoA trends are for 2004–2017. Hatching denotes statistically insignificant trends at the $2\sigma$ level.

from ACE-FTS and CLaMS for each reanalysis clearly illustrate the value of using observations when determining AoA trends, rather than relying solely on the CLaMS model and reanalysis results.

The QBO, solar, and ENSO coefficients, along with the goodness of fit ($R^2$) values, for the derived AoA anomalies are provided in the Appendix Figure A3. The QBO is a significant source of variability for AoA, particularly in the tropics. The solar cycle and ENSO have minimal impacts on stratospheric AoA. There are clearly some unaccounted sources of variability as at most 60% of the variability in AoA is explained by the regression model in the tropics, and 30% at higher latitudes. This is a source of uncertainty in the AoA trends as there could be some unknown source of decadal variability influencing the trends.

Figure 7 shows the AoA trends from 2004–2017 so that trends derived from CLaMS forced with four different reanalyses could be compared. However, results from CLaMS forced with ERA5 inputs are available to the end of 2021. AoA trends derived from these results, Figure 8, show a weakened hemispheric asymmetry: including four more years in the analysis results in a smaller rate of NH aging between 10 and 70 hPa. This shows that AoA trends in the middle stratosphere depend strongly on the period that is considered. Caution should therefore be taken when comparing the AoA trend to results from earlier studies that do not use the exact same time period for the trend calculation.

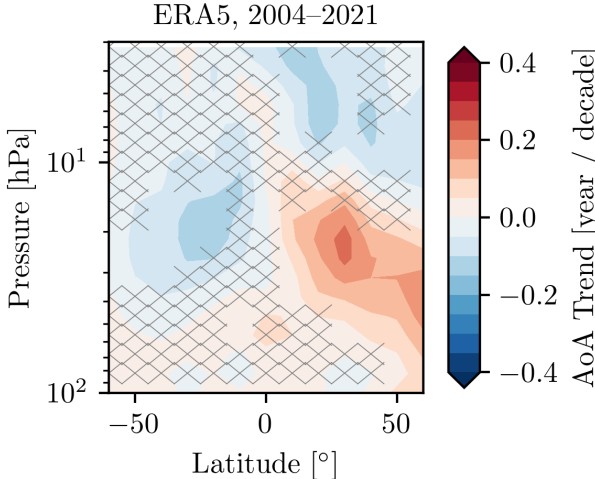

**Figure 8.** AoA trends for 2004–2021. AoA trends are derived using ACE-FTS retrievals of $N_2O$ and the relationship between AoA and $N_2O$ in CLaMS driven with ERA5. Hatching denotes statistically insignificant trends at the $2\sigma$ level.

## 4 Conclusions

Recent observations (since ∼2000) of stratospheric trace gases have opposing signs in the NH compared to the SH, which Ploeger and Garny (2022) attributed to structural changes in the BDC. One metric for quantifying the transport rate of air by the BDC, including transport from upwelling and mixing, is the AoA. Quantifying AoA trends is useful for understanding what

portion of the trace gas trends is caused by circulation changes, as opposed to changes in the source gases or chemical reaction rates. The AoA can be readily modelled, but it is more difficult to determine from observations. All existing observation-based AoA records have some limitation: AoA from MIPAS $SF_6$ is only available for 10 years, AoA from balloon observations is sparse in latitude and time (Engel et al., 2017), AoA from GOZCARDS $N_2O$ is limited in altitude (Linz et al., 2017), and AoA from NDACC HCl and $HNO_3$ is limited in both altitude and latitude (Strahan et al., 2020). Thus the goal of the present study

is to provide a new estimate of AoA and AoA trends using ACE-FTS observations, which have very good vertical resolution (1–2 km) and better latitudinal resolution (monthly global sampling) than ground-based and in-situ measurements, in addition to a nearly 20-year record.

We derived AoA trends using VMR profiles of $N_2O$, $CH_4$, and $CFC-12$ retrieved from ACE-FTS observations, along with simulations of the trace gases and AoA from CLaMS forced with four different reanalyses. The resulting AoA trends are very

similar, no matter which reanalysis is used to drive CLaMS. This is in contrast to AoA trends based only on CLaMS, which depend significantly on the choice of reanalysis used as input, highlighting the value of trace gas observations in constraining AoA trends. We found that, irrespective of which trace gas or reanalysis was used, air in the NH middle stratosphere aged by up to ∼0.3±0.1 years/decade relative to the SH over 2004–2017 near 30 hPa. The AoA interhemispheric difference trend was smaller at higher and lower levels, but remained significant between 20 hPa and 80 hPa. This trend is primarily driven by

aging in the NH, rather than a decrease in age in the SH. With ERA5 we also found that including four more years in the trend calculation, up to 2021, resulted in a smaller NH aging rate in the middle stratosphere and more equal contributions from each hemisphere to the interhemispheric difference trend.

Our finding of aging in the NH relative to the SH is consistent with earlier observational results from e.g., Strahan et al. (2020); Prignon et al. (2021); Haenel et al. (2015), but the different analysis periods used in each study makes it difficult to

directly compare the AoA trends. Strahan et al. (2020) found that the NH aged relative to the SH by 1 month/decade (0.08 years/decade) over 1994–2018 and at 52 hPa, with the majority of the trend coming from air getting younger in the SH. This is a smaller AoA interhemispheric difference trend than we find near 52 hPa for the shorter period of 2004–2017, and is also caused by changes in the opposite hemisphere. By beginning their analysis in 1994, rather than after 2000, the results of Strahan et al. (2020) were likely affected by ozone depletion. Polvani et al. (2018) showed that prior to 2000 there was a greater AoA

decrease in the SH compared to the NH due to ozone loss, and concluded that it is best to look at AoA trends before and after 2000, as is done in ozone trend studies. As our results are focused on the ozone recovery period, we see less of a reduction in the SH AoA than Strahan et al. (2020).

In general, global chemistry-climate models predict that stratospheric air will get younger throughout the 21st century, but the decrease in AoA will be small in comparison to the decrease in AoA that occurred in the second half of the 20th

century (Polvani et al., 2018; Abalos et al., 2021; Ploeger and Garny, 2022). Abalos et al. (2021) showed that AoA trends are highly sensitive to internal variability, and that at least 20 years are needed to calculate robust AoA trends. Similarly, Ploeger and Garny (2022) found a large inter-model spread in AoA trends over less than 40 years. Our results based on ACE-FTS observations are therefore not necessarily inconsistent with model predictions of air getting younger in both hemispheres as they are only based on 13 years (and in one case 17 years) of measurements. The smaller aging trend in the NH middle

stratosphere that we find when AoA trends are calculated over 2004–2021 instead of 2004–2017 provides some evidence that the NH middle stratosphere aging is only a feature of the short observation period and likely related to natural variability, and that the agreement between observations and models can improve with a longer data record.

*Code and data availability.* ACE-FTS data are available by registration at https://databace.scisat.ca/level2/ (ACE-FTS, 2022).

ACE-FTS data quality flags are available from https://doi.org/10.5683/SP2/BC4AT (Sheese and Walker, 2022).

The NOAA/GML HATS $N_2O$ is available at https://gml.noaa.gov/hats/combined/N2O.html (NOAA/GML, 2022).

The LOTUS regression code and documentation is available at https://arg.usask.ca/docs/LOTUS_regression/index.html (Damadeo et al., 2022).

The monthly zonal mean ACE-FTS AoA anomalies derived from $N_2O$ are available at https://doi.org/10.5281/zenodo.11492264 (Dubé et al., 2024).

The CLaMS model results are available upon request from Felix Ploeger (f.ploeger@fz-juelich.de).

Example code for producing the results presented in the manuscript is available at https://zenodo.org/records/13952202 (Dubé, 2024).

# Appendix A: Extra Figures

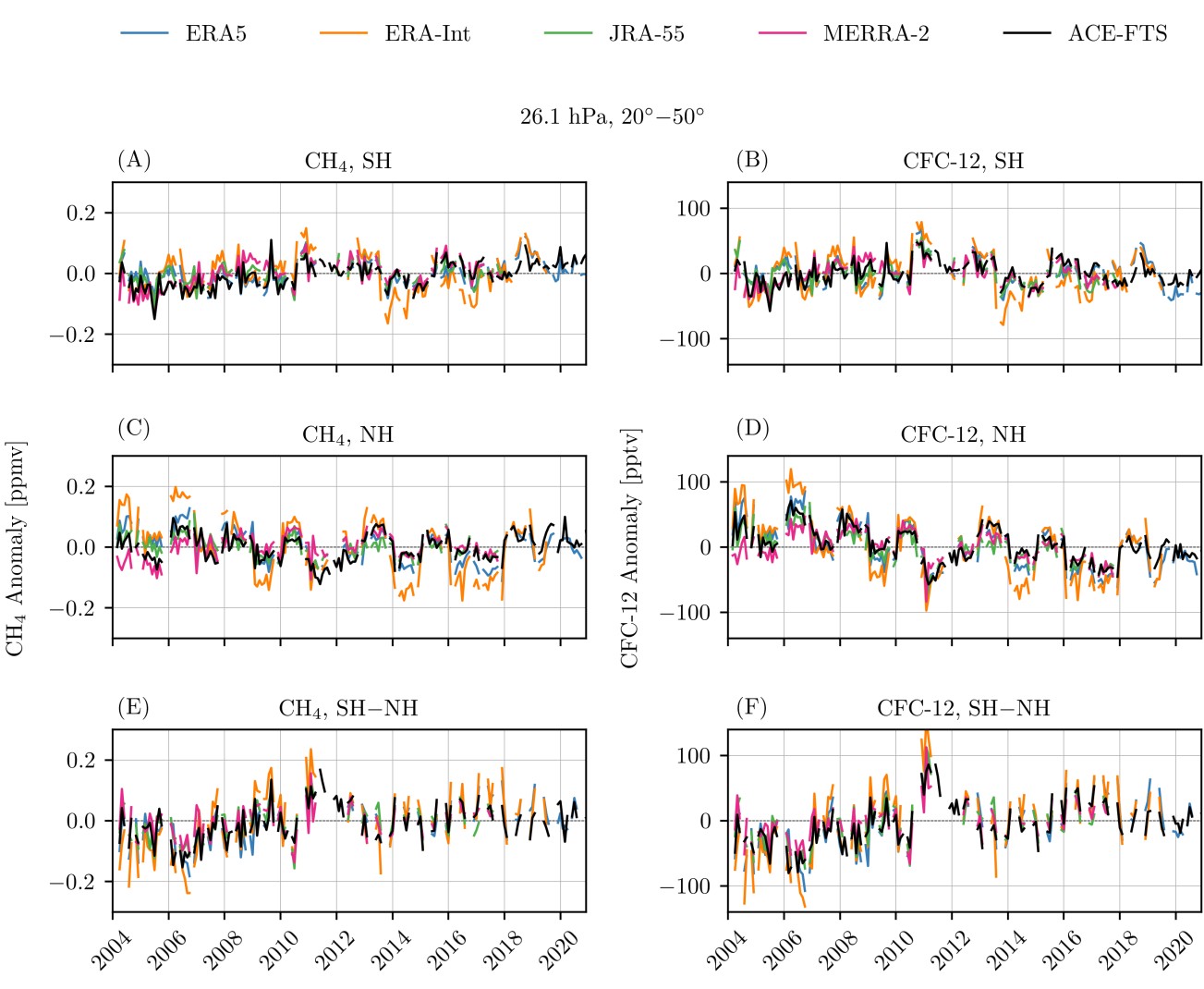

**Figure A1.** Left: Deseasonalized monthly zonal mean anomaly time series at 26.1 hPa for ACE-FTS $CH_4$ and $CH_4$ from CLaMS driven with four different reanalyses. Right: Deseasonalized monthly zonal mean anomaly time series for ACE-FTS $CFC-12$ and $CFC-12$ from CLaMS driven with four different reanalyses. Panels are divided into SH (top row), NH (centre row), and SH−NH difference (bottom row).

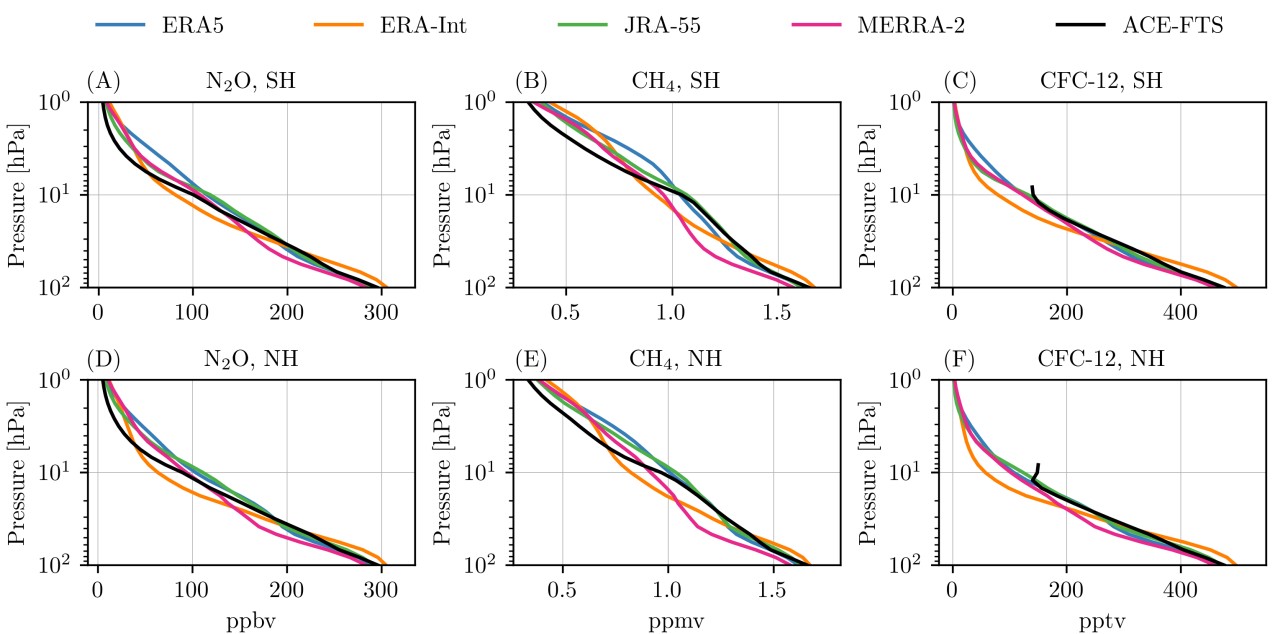

**Figure A2.** Mean profiles of $N_2O$, $CH_4$, and CFC$-12$ from ACE-FTS and CLaMS driven with four different reanalyses over 2004–2017. The profiles are separated by hemisphere: profiles in the top row are for 20°N-50°N, and profiles in the bottom row are for 50°S-20°S.

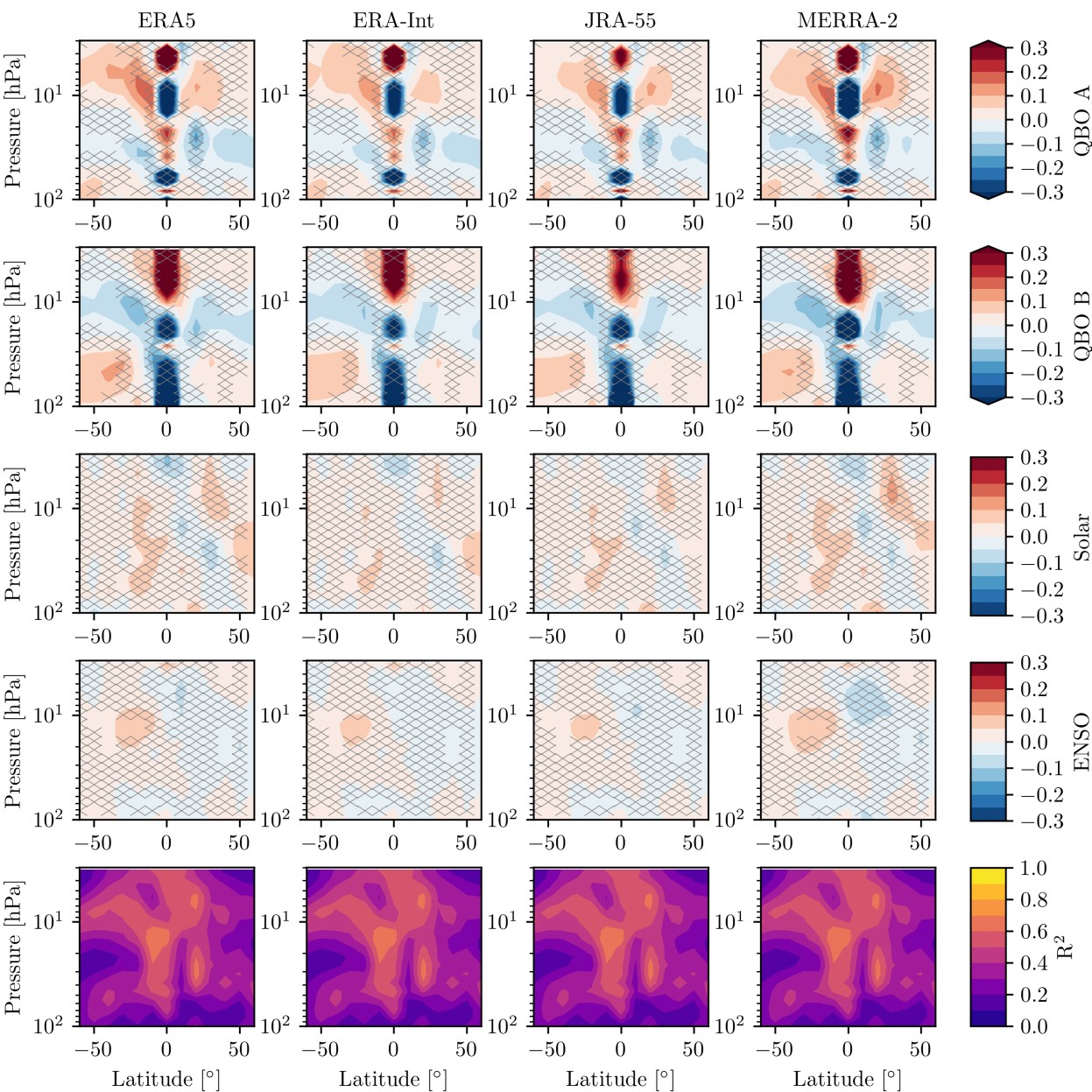

**Figure A3.** Rows 1–4: Coefficients for a multiple linear regression fit to AoA from 2004–2017. Hatching denotes statistically insignificant values at the $2\sigma$ level. Row 5: $R^2$ values for the regression. Results are shown for AoA derived from ACE-FTS $N_2O$ retrievals and from AoA and $N_2O$ from CLaMS driven with four different reanalyses.

*Author contributions.* KD performed the analysis and wrote the manuscript, with input from all co-authors. ST conceptualized and supervised the project. FP provided the CLaMS simulations. KW provided advice on the ACE-FTS data.

*Competing interests.* We declare that none of the authors have any competing interests.

*Acknowledgements.* This research has been supported by the Canadian Space Agency (grant no. 21SUASULSO). The Atmospheric Chemistry Experiment (ACE) is a Canadian-led mission mainly supported by the CSA and the NSERC, and Peter Bernath is the principal investigator.

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
