# Peer review of "Hemispheric asymmetry in recent stratospheric age of air changes"

_EGUsphere, 2024_

## Referee Comment (RC1)

Review of the manuscript

**Hemispheric asymmetry in recent stratospheric age of air changes**

submitted to ACP by K. Dubé et al.

S. Chabrillat, BIRA-IASB, August 2024

**General Comments**

This study is a timely and useful contribution on the long-term changes in the BDC and their hemispheric asymmetry, a topic which is currently much discussed in the middle atmosphere community.

I found the text very well written and pleasant to read, except for the description of the derivation methods (see MC1 and SC1 below). I also feel that the submitted manuscript misses an important opportunity, as the discussion of ACE $N_2O$-derived AoA trends (section 3.2) does not explicitly compare the results with those presented in the modelling study which is the steppingstone for this paper (i.e. Ploeger and Garny, 2022): see MC2 below.

Since they do not imply any new model experiment or additional dataset, all suggested revisions are minor. Once they are addressed, I wholeheartedly recommend publication – preferably in *Atmos. Chem. Phys.*

**Major comments**

**MC1.** The derivations of trends in AoA interhemispheric differences (section 3.1) and of latitude-dependent AoA derived from ACE-FTS retrievals of $N_2O$ (section 3.2) both rely on the preliminary derivation of the linear regression coefficients (the "slope") between model-derived AoA and model-derived relative anomalies of $N_2O$ (also done with $CH_4$ and CFC-12 in section 3.1). The derivation of these slopes is thus the key point of the methodology, and all studies aiming to derive AoA from observations of long-lived gases are expected to rely on similar methods. Hence it is important to describe this step in a clear and complete manner, which is currently not the case.

Rather than a cursory description aimed at explaining Figure 3, this should be expanded into one or two paragraphs and an additional figure to illustrate the derivation prior to showing its outcome. Focusing on the $N_2O$ case, I would expect such a figure to show a scatter plot with $N_2O$ relative anomalies on the X-axis and modelled AoA anomalies on the Y-axis, as well as the least-squares fit by a linear function (or better, three fits: one with the derived slope and two to show its $\pm 2\sigma$ uncertainty). Or maybe the axes of such a plot should rather be the *SH-NH differences* of the $N_2O$ relative anomalies and AoA anomalies?

The methodology description in section 3.1 should also clearly state when the SH-NH differences (of $N_2O$ and AoA) are computed: does this happen before or after the derivation of *relative* anomalies of $N_2O$/$CH_4$/CFC12 vmr?

The relative anomaly is itself defined (line 161) as "*the anomaly divided by the* [its] *overall mean*". I understand that these overall means have one single value for each species, pressure level and reanalysis, either for each hemisphere separately or directly for the interhemispheric differences. If that is correct, consider adding a figure in the Appendix to show the vertical profiles of these overall means. If this is not correct, please clarify.

**MC2.**   This study is built on prior CLaMS simulations of AoA and long-lived tracers using four different reanalyses (Ploeger and Garny, 2022). This earlier paper showed latitude-pressure distributions of AoA trends (Fig.1a-1d in P&G2022), confirming that such CTM results are very dependent on their input reanalysis (Chabrillat et al., 2018). Hence, even though CTMs use as input observation-derived reanalyses of stratospheric dynamics, they cannot be used to unambiguously evaluate the ability of pure models (i.e. GCCM) to simulate the past evolution of the BDC. As explained in the introduction of the present manuscript, this is the motivation to derive AoA trends by combining CTM results with satellite-based observations of long-lived tracers: here ACE-FTS observations of $N_2O$ are scaled by model-derived ratios ("slopes") representing the linear relationship between AoA and the tracer, as a function of latitude and pressure level, and this scaling is appropriately applied prior to the derivation of the trends.

The final figure shows very well that such "hybrid" derivations of AoA trends do not depend much on the input reanalysis used by the CTM, *thanks to* the injection of observations of long-lived tracers. Yet the final figures and their discussion (lines 198-233) fail entirely to mention the role played by observations. While this role is explained in prior sections and repeated in the conclusion, I still think that the discussion could be much improved by highlighting this point. Specific comments SC13-SC15 suggest some steps in that direction, especially SC15 to expand the last figure with CTM results.

**Specific Comments**

Original text is copied in italics, suggestions for corrections are typed in bold.

**SC1.**   The words "*ACE-FTS gases*" are used throughout the text, with "gases" as a shorthand for "long-lived gas-phase tracers". This is not good wording because these "gases" are material substances, while you are manipulating numerical quantities which describe the variations in the abundances of these gases. To stick with rigorous vocabulary for remote-sensing measurements: the actual ACE-FTS "observations" are spectra of solar light partially absorbed by the atmosphere, while the volume mixing ratio (vmr) profiles which are delivered by the ACE-FTS team are "retrievals" from these observations. The text will become much clearer if you replace all relevant instances of "gases" by the quantities actually retrieved from ACE-FTS observations or derived from these retrievals, i.e. "**ACE-FTS vmr**" (volume mixing ratios) for the natively retrieved quantities or "**ACE-FTS (relative) vmr anomalies**" for the quantities which you derive from them. You will only need to define "vmr" once in section 2 or in the introduction of section 3.

**SC2.**   The introduction is very well written. It only lacks an (often overlooked) reference to Garcia et al. (2011) which explains fundamental difficulties and limitations in deriving significant AoA trends from observations of trace gas species.

**SC3.** Line 29: "*The BDC is the mechanism for* **long-range** *mass transport within the stratosphere*".

**SC4.** Lines 42-43: reanalyses are *not* weather forecast models. They are datasets with high vertical and horizontal resolution providing, every few hours over several decades, physically consistent snapshots of atmospheric wind fields and temperature (and many other parameters not used here). These datasets are generated by NWP assimilation systems which combine weather model forecasts with many sources of homogenized observations. This is a good place to cite the S-RIP, either by citing its introductory paper (Fujiwara et al., 2017), the corresponding ACP/ESSD issue, or the whole S-RIP report.

**SC5.** Line 97: please provide some details about the initial states of these four simulations. When do they start? Do they all use the same initial conditions?

**SC6.** Lines 15-118: are these NOAA/GML in-situ observations are collected at the surface rather or the free troposphere? Please clarify.

**SC7.** Line 122: "*…and interpolating the CLaMS output to the representative ACE-FTS profile locations and times at 30 km*". Please clarify, e.g. "*…and interpolating the CLaMS output to a location and time representative of the whole ACE-FTS profiles, using their locations and times at 30 km*".

**SC8.** Figure 1: it is difficult to distinguish the ACE-FTS lines from the MERRA-2 lines (and probably not readable by color-blind readers but I do not know how to fix this). Please consider plotting ACE-FTS in black or grey, which is a common choice for observational datasets.

**SC9.** Lines 134-135: "*there*  **are** *clear difference***s** *between the datasets when considering changes over the*  **shared** *17-year period.*".

**SC10.** Lines 135-139: the year 2004 is not a very good exemple, as SH-NH AoA (fig. 1F) still differs for this initial year (e.g. anomaly of -1year with MERRA-2 versus ~0.7year with ERA-Int). Maybe use instead 2005 or 2006?

**SC11.** Line 150 and legend of Fig. 2: "*...ACE-FTS CFC−12 observations are only available up to ∼8 hPa.*" It should still be possible to show the results for model-only correlations above 8 hPa (i.e. in panel C) ?

**SC12.** Line 188: as written, the sentence makes no sense. It is not possible to subtract "emission trends" from "ACE-FTS N2O observations" as these quantities do not have the same dimensions (strictly speaking, emission trends are mass/surface/time/time). Since you describe the actual procedure in the next paragraph, you could simply replace this sentence by a more general one introducing the next paragraph, e.g. "**We give here a simple procedure to remove the effect of emission trends on trends of stratospheric $N_2O$**. *This cannot…*"

**SC13.** The legends of Figs 5 and 6 should be clarified as follows:

> *Figure 5/6. AoA anomaly/trends* **derived from ACE-FTS retrievals of $N_2O$**. *Results are shown*  **using** *CLaMS driven with four different reanalyses*. […]

**SC14.** Line 217: "*The positive AoA trend in the NH is consistent with reanalysis results from Monge-Sanz et al. (2022)*". The cited reference is the chapter 5 of S-RIP report, which compiles many different papers discussing BDC trends using reanalyses. What specific result/figure are you referring? Please add a citation to the corresponding paper and develop this part of the discussion.

**SC15.** I strongly suggest expanding Figure 6 by inserting a first row showing the corresponding "pure" CLaMS results, i.e. a repetition of Figs 1a-1d in P&G2022. The AoA trends should of course be re-computed using the same 2004-2017 period and MLR processing as used here, and the contour plots should keep the same, simpler design as in Fig. 6 (no potential temperature nor zonal wind contours; WMO tropopause very useful; hatching quite necessary). This would allow an easy comparison between earlier CLaMS results and the current "hybrid" results and could allow a quite fruitful discussion.

**SC16.** Line 263: "*In general,* **global chemistry-climate** *models predict that…*"

**Typos, wording etc.**

I did not find any: congratulations!

**Additional bibliographical references**

Fujiwara, M., Wright, J. S., Manney, G. L., Gray, L. J., Anstey, J., Birner, T., Davis, S., Gerber, E. P., Harvey, V. L., Hegglin, M. I., Homeyer, C. R., Knox, J. A., Krüger, K., Lambert, A., Long, C. S., Martineau, P., Molod, A., Monge-Sanz, B. M., Santee, M. L., Tegtmeier, S., Chabrillat, S., Tan, D. G. H., Jackson, D. R., Polavarapu, S., Compo, G. P., Dragani, R., Ebisuzaki, W., Harada, Y., Kobayashi, C., McCarty, W., Onogi, K., Pawson, S., Simmons, A., Wargan, K., Whitaker, J. S., and Zou, C.-Z.: Introduction to the SPARC Reanalysis Intercomparison Project (S-RIP) and overview of the reanalysis systems, Atmos. Chem. Phys., 17, 1417–1452, https://doi.org/10.5194/acp-17-1417-2017, 2017.

Garcia, R. R., Randel, W. J., and Kinnison, D. E.: On the Determination of Age of Air Trends from Atmospheric Trace Species, J. Atmos. Sci., 68, 139–154, https://doi.org/10.1175/2010JAS3527.1, 2011.

---

## Author Comment (AC1)

**Response to Reviewers: Hemispheric asymmetry in recent stratospheric age of air changes**

We thank both reviewers for their feedback on our manuscript! Responses to specific comments are provided below in blue text.

**Reviewer 1: S. Chabrillat**

**MC1:** The derivations of trends in AoA interhemispheric differences (section 3.1) and of latitude dependent AoA derived from ACE-FTS retrievals of N2O (section 3.2) both rely on the preliminary derivation of the linear regression coefficients (the "slope") between model-derived AoA and model derived relative anomalies of N2O (also done with CH4 and CFC-12 in section 3.1). The derivation of these slopes is thus the key point of the methodology, and all studies aiming to derive AoA from observations of long-lived gases are expected to rely on similar methods. Hence it is important to describe this step in a clear and complete manner, which is currently not the case. Rather than a cursory description aimed at explaining Figure 3, this should be expanded into one or two paragraphs and an additional figure to illustrate the derivation prior to showing its outcome. Focusing on the N2O case, I would expect such a figure to show a scatter plot with N2O relative anomalies on the X-axis and modelled AoA anomalies on the Y-axis, as well as the least-squares fit by a linear function (or better, three fits: one with the derived slope and two to show its ±2s uncertainty). Or maybe the axes of such a plot should rather be the SH-NH differences of the N2O relative anomalies and AoA anomalies? The methodology description in section 3.1 should also clearly state when the SH-NH differences (of N2O and AoA) are computed: does this happen before or after the derivation of relative anomalies of N2O/CH4/CFC12 vmr?

The relative anomaly is itself defined (line 161) as "the anomaly divided by the [its] overall mean". I understand that these overall means have one single value for each species, pressure level and reanalysis, either for each hemisphere separately or directly for the interhemispheric differences. If that is correct, consider adding a figure in the Appendix to show the vertical profiles of these overall means. If this is not correct, please clarify.

Thank you for the comment. The slope calculation is indeed an important part of the analysis and it needs to be explained properly. We have added a figure to the manuscript that shows the scatter plots of AoA vs. each of N2O, CH4, and CFC-12 for each reanalysis and for two sample pressure levels. The plots are in terms of the SH-NH difference. The slopes of the fit lines in the scatter plots correspond to the slope values that we use to

calculate the age of air trends. The calculation of this fit line is done using the standard least-squares method, so we do not believe that any further details are required.

[Figure]

The SH-NH difference is done after finding the anomalies for each hemisphere. We have added text clarifying this to Section 3.1.

Your understanding of the relative anomaly is correct. A figure showing the mean profiles that we divide by to get the relative anomaly has been added to the Appendix. The figure is also included here:

[Figure]

**MC2:** This study is built on prior CLaMS simulations of AoA and long-lived tracers using four different reanalyses (Ploeger and Garny, 2022). This earlier paper showed latitude-pressure distributions of AoA trends (Fig.1a-1d in P&G2022), confirming that such CTM results are very dependent on their input reanalysis (Chabrillat et al., 2018). Hence, even though CTMs use as input observation-derived reanalyses of stratospheric dynamics, they cannot be used to unambiguously evaluate the ability of pure models (i.e. GCCM) to simulate the past evolution of the BDC. As explained in the introduction of the present manuscript, this is the motivation to derive AoA trends by combining CTM results with satellite-based observations of long-lived tracers: here ACE-FTS observations of N2O are scaled by model-derived ratios ("slopes") representing the linear relationship between AoA and the tracer, as a function of latitude and pressure level, and this scaling is appropriately applied prior to the derivation of the trends. The final figure shows very well that such "hybrid" derivations of AoA trends do not depend much on the input reanalysis used by the CTM, thanks to the injection of observations of long-lived tracers. Yet the final figures and their discussion (lines 198-233) fail entirely to mention the role played by observations. While this role is explained in prior sections and repeated in the conclusion, I still think that the discussion could be much improved by highlighting this point. Specific comments SC13-SC15 suggest some steps in that direction, especially SC15 to expand the last figure with CTM results.

Thank you, we agree that explaining the role of the ACE-FTS observations is important and have elaborated on this in the text. Your suggestion in SC15 to show the CLaMS-only AoA trends is a very good idea as including these results clearly highlights how different the AoA

trends are when the ACE-FTS observations are not considered. We have added this to the figure, along with a discussion, in Section 3.2.

**SC1:** The words "ACE-FTS gases" are used throughout the text, with "gases" as a shorthand for "long-lived gas-phase tracers". This is not good wording because these "gases" are material substances, while you are manipulating numerical quantities which describe the variations in the abundances of these gases. To stick with rigorous vocabulary for remote-sensing measurements: the actual ACE-FTS "observations" are spectra of solar light partially absorbed by the atmosphere, while the volume mixing ratio (vmr) profiles which are delivered by the ACE-FTS team are "retrievals" from these observations. The text will become much clearer if you replace all relevant instances of "gases" by the quantities actually retrieved from ACE-FTS observations or derived from these retrievals, i.e. "ACE-FTS vmr" (volume mixing ratios) for the natively retrieved quantities or "ACE-FTS (relative) vmr anomalies" for the quantities which you derive from them. You will only need to define "vmr" once in section 2 or in the introduction of section 3.

Thank you for the recommendation, the text has been adjusted accordingly, and Section 2 now mentions that VMR profiles are retrieved from the ACE observations.

**SC2:** The introduction is very well written. It only lacks an (often overlooked) reference to Garcia et al. (2011) which explains fundamental difficulties and limitations in deriving significant AoA trends from observations of trace gas species.

This reference has been added to the introduction.

**SC3**: Line 29: "The BDC is the mechanism for long-range mass transport within the stratosphere"

 Done

**SC4:** Lines 42-43: reanalyses are not weather forecast models. They are datasets with high vertical and horizontal resolution providing, every few hours over several decades, physically consistent snapshots of atmospheric wind fields and temperature (and many other parameters not used here). These datasets are generated by NWP assimilation systems which combine weather model forecasts with many sources of homogenized observations. This is a good place to cite the S-RIP, either by citing its introductory paper (Fujiwara et al., 2017), the corresponding ACP/ESSD issue, or the whole S-RIP report

Thank you for pointing this out, the explanation of reanalyses has been changed: "Reanalyses are datasets with high vertical and horizontal resolution that are generated by assimilation systems which combine global weather forecast models and input

observations from many sources to provide a physically consistent estimate of the past atmospheric state (Fujiwara et al., 2017)."

**SC5:** Line 97: please provide some details about the initial states of these four simulations. When do they start? Do they all use the same initial conditions?

The simulations begin in 1979, except the MERRA2 simulation which begins in 1980 (due to data availability). The same initial conditions for trace gases and age of air for all four simulations are taken from climatological data (see Pommrich et al., 2014, GMD). For each case, there is a time-slice spin-up simulation of 10 years by repeating the first year ten times, before the transient simulation starts (see Ploeger et al., 2021, ACP). Spin-up effects are not important for our study as we only use results from 2004 onwards.

Pommrich, R., Müller, R., Grooß, J. U., Konopka, P., Ploeger, F., Vogel, B., ... & Riese, M. (2014). Tropical troposphere to stratosphere transport of carbon monoxide and long-lived trace species in the Chemical Lagrangian Model of the Stratosphere (CLaMS). Geoscientific model development, 7(6), 2895-2916.

Ploeger, F., Diallo, M., Charlesworth, E., Konopka, P., Legras, B., Laube, J. C., ... & Riese, M. (2021). The stratospheric Brewer–Dobson circulation inferred from age of air in the ERA5 reanalysis. Atmospheric chemistry and physics, 21(11), 8393-8412.

**SC6:** Lines 15-118: are these NOAA/GML in-situ observations are collected at the surface rather or the free troposphere? Please clarify.

The measurements are made at the surface, this is now specified in the manuscript.

**SC7:** Line 122: "...and interpolating the CLaMS output to the representative ACE-FTS profile locations and times at 30 km". Please clarify, e.g. "...and interpolating the CLaMS output to a location and time representative of the whole ACE-FTS profiles, using their locations and times at 30 km".

The sentence has been clarified.

**SC8:** Figure 1: it is difficult to distinguish the ACE-FTS lines from the MERRA-2 lines (and probably not readable by color-blind readers but I do not know how to fix this). Please consider plotting ACEFTS in black or grey, which is a common choice for observational datasets.

The ACE-FTS have been changed to black in both Figure 1 and Figure A2 (previously Figure A1).

**SC9:** Lines 134-135: "there is a are clear differences between the datasets when considering changes over the full shared 17-year period.".

Fixed

**SC10:** Lines 135-139: the year 2004 is not a very good example, as SH-NH AoA (fig. 1F) still differs for this initial year (e.g. anomaly of -1year with MERRA-2 versus ~0.7year with ERA-Int). Maybe use instead 2005 or 2006?

We have changed the text to use 2006 as an example, instead of 2004.

**SC11:** Line 150 and legend of Fig. 2: "...ACE-FTS CFC−12 observations are only available up to ~8 hPa." It should still be possible to show the results for model-only correlations above 8 hPa (i.e. in panel C) ?

You are correct, panel C of the plot now shows the model-only correlations up to 1 hPa.

**SC12:** Line 188: as written, the sentence makes no sense. It is not possible to subtract "emission trends" from "ACE-FTS N2O observations" as these quantities do not have the same dimensions (strictly speaking, emission trends are mass/surface/time/time). Since you describe the actual procedure in the next paragraph, you could simply replace this sentence by a more general one introducing the next paragraph, e.g. "We give here a simple procedure to remove the effect of emission trends on trends of stratospheric N2O. This cannot…"

This sentence has been removed.

**SC13:** The legends of Figs 5 and 6 should be clarified as follows: Figure 5/6. AoA anomaly/trends derived from ACE-FTS retrievals of N2O. Results are shown for using CLaMS driven with four different reanalyses. […]

The captions in what are now Figs. 6,7,8 have been clarified.

**SC14:** Line 217: "The positive AoA trend in the NH is consistent with reanalysis results from MongeSanz et al. (2022)". The cited reference is the chapter 5 of S-RIP report, which compiles many different papers discussing BDC trends using reanalyses. What specific result/figure are you referring? Please add a citation to the corresponding paper and develop this part of the discussion.

We have removed this citation and instead reference Ploeger and Garny (2022) as the focus is on comparing with those results.

**SC15:** I strongly suggest expanding Figure 6 by inserting a first row showing the corresponding "pure" CLaMS results, i.e. a repetition of Figs 1a-1d in P&G2022. The AoA trends should of course be recomputed using the same 2004-2017 period and MLR processing as used here, and the contour plots should keep the same, simpler design as in Fig. 6 (no potential temperature nor zonal wind contours; WMO tropopause very useful; hatching quite necessary). This would allow an easy comparison between earlier CLaMS results and the current "hybrid" results and could allow a quite fruitful discussion.

This is a very good idea, thanks! We have added the CLaMS-only AoA trends to the figure:

[Figure]

**SC16:** Line 263: "In general, global chemistry-climate models predict that..."

Done.

**Reviewer 2: T. Wagenhäuser**

**Regarding Major comments from RC1**

Regarding MC1 form RC1: I agree that describing the derivation of the slopes is an important part of the manuscript, that should be addressed more clearly in a way to facilitate reader understanding and potentially reproducing the results. However, if the authors consider the details of this methodology to be well-established and not central to reproducing their results, they could provide a more detailed description in the supplementary material.

We have elaborated upon the methodology in the manuscript, see response to Reviewer 1.

Regarding MC2 from RC2: I agree that this manuscript could benefit from highlighting the differences between prior model-based results and this hybrid approach, especially by drawing a more explicit comparison with the previous study from Ploeger and Garny (2022).

Yes, we agree that this is a very good suggestion. See response to Reviewer 1.

**SC1:** Line 108,109: From my understanding, only the trend in AoA interhemispheric difference is calculated from all three species mentioned, whereas the latitudinally resolved AoA trend is derived only from N2O. A more general phrasing such as "We derive trends in both the AoA interhemispheric difference (SH-NH) and the latitudinally resolved AoA from trace gas-AoA correlations.", may suffice, as the specific substances and their roles are specified further down.

You are correct, the sentence has been changed.

**SC2:** Figure 1 caption: Please consider aligning the figure caption more closely with the information provided in the main text to enhance clarity and make it more self-descriptive. In particular, adopting the term "monthly zonal mean", as used in the main text, would make the figure caption more accurate.

The figure caption now specifies that the monthly zonal mean anomaly is shown.

**SC3:** Figure 1: In the main text the magnitude of differences across the reanalyses is discussed (l. 134 – 139). To complement this discussion, you may consider adding a figure showing a box plot time series (one box plot per year, each box based on 5 values (reanalyses and observations annual mean)), which might help highlight periods of low or high agreement between different reanalyses and observations.

Thank you for the suggestion. We made the box plot for the AoA anomaly (see below) but decided that it does not contain any information that is not already in Figure 1. We did also consider replacing Figure 1 with the box plot version, but then it is not possible to see the very good agreement in the short-term variability amongst all the datasets.

[Figure]

**SC4:** Figure 2 caption: I suggest further elaborating on the figure caption to make it more self-descriptive. In particular, the caption should state that the correlation between AoA and the three trace gases is shown and that atmospheric pressure is used to draw vertical profiles.

 More details have been added to the caption of Figure 2.

**SC5:** Figure 3 caption: Again I think that you should add information for clarity. E.g. you could use "top row", "middle row" and "bottom row" to avoid ambiguities. Also you could add "at different pressure levels" or something similar to the first sentence of the caption.

Further detail has been added to the caption.

**SC6:** Figure 4 shows that the AoA anomalies are consistent across different input reanalyses. The manuscript could benefit from a clearer discussion of how similar the results are across reanalyses, ideally with metrics to quantify this agreement (such as correlation coefficients).

Do you mean Figure 5? The correlation is essentially perfect as all datasets are just a scaled version of the ACE-FTS N2O relative anomaly and the scale factors are constant in

time for a given latitude and altitude. We have added a sentence mentioning this to the manuscript:

"The AoA anomalies are very similar for all four reanalyses, and the short-term variability in the AoA from all four reanalyses is identical since all cases are a scaled version of the ACE-FTS N2O relative anomaly and the scale factors do not vary in time."

**Technical suggestions**

- 225: "some unaccounted sources [...]" without the "for" for better readability
  - o Done
- You might consider uploading your code used to derive your results, ideally in the form of a Jupyter notebook, to enhance reproducibility and future application.
  - o Thank you for the suggestion, a Jupyter notebook has been uploaded to zenodo: https://zenodo.org/records/13952202

---

## Author Response (AR2)

**Response to Editor: Hemispheric asymmetry in recent stratospheric age of air changes**

Thank you for the suggestions, we have implemented the changes, as described here in blue text.

L19: Please consider alternative phrasing to avoid two parenthesis sets together "... (BDC) (e.g., Stiller et al., 2017; Bonisch et al., 2011; Fu et al., 2019)"

It is difficult to reword the sentence in this case as it is necessary to both define the BDC acronym and provide references. We have instead changed the text so that the acronym definition and the references are in the same set of parentheses: "[...]an altered Brewer-Dobson Circulation (BDC, e.g., Stiller et al., 2017; Bönisch et al., 2011; Fu et al., 2019), amongst others."

L46: Please consider alternative phrasing to avoid two parenthesis sets together "... (CTM) (Chipperfield, 2006)". Please consider alternative reference as this one is used in the immediately following sentence. I suggest Monge-Sanz et al. (2007), which used the same CTM and focused on comparing different reanalyses, instead of comparing model configurations.

The sentence has been changed to "A Chemical Transport Model (CTM) can be used to calculate AoA using reanalysis fields as input (e.g., Monge-Sanz et al., 2007)."

L 108: Please add "in CLaMS simulations" --> "..for each reanalysis in CLaMS simulations .."

The sentence has been changed to "The same CLaMS simulations for each reanalysis are used here."

P 10, Table 1: Please Include in the caption the fact that the results shown here come from the average of the four runs driven by the corresponding four reanalyses. This is specified in the main text, it will be clearer to have it here as well.

The caption has been changed to "Trends in AoA interhemispheric difference (SH−NH) for 2004–2017. Values are the mean AoA trend derived from ACE-FTS and four CLaMS runs, each using input from a different reanalysis. Trend units are year/decade. Errors are the 2σ uncertainty."

P 11, Figure 5: Please add to the caption that the hatching for those two reanalyses is found at the top levels at the highest latitudes shown (they are difficult to spot).

The caption now specifies that the hatching is at the highest latitude and pressure levels.

L 236: Please delete "for" and replace "at latitudes >−40∘" with "at latitudes larger than 40∘S"

The sentence has been changed to "The AoA trend in the SH is largely insignificant, except between 10 hPa and 30 hPa at latitudes larger than 40∘S, where the air is getting younger by up to 0.2 years/decade."

L 244: Correct typo "are"

Done

L245: Delete "do"

Done

L 251: Please add one sentence clarifying that these differences when adding ACE-FTS information specifically apply to CLaMS, not necessarily to all CTMs. Please also add a brief mention about the overall agreement between your ACE+ERA-Int results in Fig.7 and those from Fig. 5.38 in Monge-Sanz et al. (2022).

We have specified that are results are for the CLaMS model: "The dissimilarity between the CLaMS-only AoA trends for each reanalysis, compared to the similarity of the AoA trends derived from ACE-FTS and CLaMS for each reanalysis clearly illustrate the value of using observations when determining AoA trends, rather than relying solely on the CLaMS model and reanalysis results."

Comparison to Fig. 5.38: As our manuscript discusses the large dependence of the AoA trends on the time period considered in the analysis, we find it difficult to directly compare our results to those presented in Fig. 5.38 of Monge-Sanz et al. (2022), as that figure shows trends for 2003-2011 (and is based on the TOMCAT CTM). We instead include the following sentence:

"The differences between AoA trends derived from a CTM forced with different reanalyses are also apparent in other time periods and for CTMs besides CLaMS, as demonstrated by Monge-Sanz et al. (2022)."

L 284: Please specify that this is for ERA5 --> "With ERA5 we also found that…"

Done

L 442: Please also add the url identifier for this reference:
https://orfeo.belnet.be/handle/internal/9819

This URL has been added to the bibtex file. Note that it does not appear in the compiled pdf with the current settings on the latex template- the copyeditor for the journal will have to make sure that the URL appears in the paper once it has received the final formatting.